# Using Import Data to Predict the Potential of Introduction of Alert Alien Species to South Korea

**Aram Jo [1], Seunghun Son [2] and Dongeon Kim [1,\*]**

[1] Division of Ecological Threat Management, National Institute of Ecology, Seocheon 33657, Korea
[2] Biological Resources Research Department, National Institute of Biological Resources, Incheon 22689, Korea
**\*** Correspondence: eco0106@nie.re.kr; Tel.: +82-041-950-5803

**Abstract:** As globalization progresses, human activities, such as travel and trade, are rapidly increasing beyond national boundaries. It is increasingly recognized that places, such as ports and airports, where trade occurs play a major role as an introduction pathway for alien species. In this study, we focused on evaluating the possibility of introduction of Alert Alien Species (AAS) through trade data among countries. The natural and distribution range of AAS were analyzed along with import data by country. There were large differences between the number of AAS distributed in a country and the import weight of items related to the import of AAS from the country. Fish, which account for 76% of the import weight of AAS, 43 and 40 species of the 84 species of AAS were distributed in US and Russia, respectively. However, the import weight of items related to the import of fish designated as AAS from these countries were extremely low. This finding suggests that trade, which is the main introduction pathway, is not taken into account in the designation of AAS. For future management plans for non-introduced alien species, species with a high possibility of introduction into South Korea through trade should be prioritized using import data.

**Keywords:** invasive alien species; biological invasion; trade; pathway of introduction

## 1. Introduction

Starting from mass migrations of species during the Middles Ages in the 1500s when Europeans moved to North America to the increased trade in the 1800s during the Industrial Revolution that has continued into the current era of globalization, species' boundaries between countries are gradually disappearing [1]. As a result, this has caused an increase in alien species moving away from their natural range and resulted in damaged ecosystems, hybridization, and competition with native species, which has thereby led to a decline in biodiversity and economic value [2,3]. In South Korea, there has been serious damage caused by invasive alien species such as the spotted lanternfly (*Lycorma delicatula*), nutria (*Myocastor coypus*), and common cordgrass (*Spartina anglica*) [4]. The economic costs of such invasive alien species from 1970 to 2017 have been estimated to be at a minimum of USD 1.28 trillion [5].

According to the 2nd Management Plan of Alien Species (2019–2023) [4], the confirmed number of alien species introduced into South Korea was 1109 species in 2011, which more than doubled to 2160 species in 2018. It is assumed, considering cases of unclear introduction pathway or deliberately unreported import, that many more alien species have been introduced domestically. Accordingly, the Ministry of Environment designated non-introduced alien species with a potential risk of causing harm to the ecosystem if they are introduced in South Korea as "alert alien species (AAS)" to be managed as part of the Act on the Conservation and Use of Biological Diversity (hereafter, the "Biodiversity Act") [2]. The standards for AAS designation are as follows: 1. species whose risks are recognized internationally; 2. species known to have caused social or ecological damage;

3. species which have similar ecological or genetic characteristic to ecosystem-disturbing species or species with potential risks to the ecosystem; 4. species that have a high possibility of establishment in South Korea because their natural habitat conditions are similar to the environment of South Korea. 5. species known to impact human health or spread various diseases [6]. For AAS, risk assessment must be conducted when they are imported or introduced (Article 22 of the Biodiversity Act). Risk assessment involves evaluating the possibility of introduction, establishment, and spread of the subject species and their impact on ecosystems, society, and the economy (Article 21 (2) of the Biodiversity Act). As of April 2021, 300 species have been designated as AAS [7]. The Ministry of Environment plans to expand the number of AAS to 1000 by [8].

Despite such legal efforts, preventing the introduction and spread of alien species remains highly challenging. Biological invasion, which is the process of species becoming invasive in another habitat away from their original habitat, involves various components such as introduction, establishment, and spread. However, most studies on AAS in South Korea are about the spread and establishment possibility when they are introduced rather than the introduction possibility [9,10].

Alien species are typically introduced into new habitats by human activities such as travel, transport, and trade. Among them, trade is very closely associated with biological invasion [1,11,12]. First, alien species can be introduced unintentionally, such as attached to ships or mixed with other products. *Solenopsis invicta*, designated among 100 of the world's worst invasive alien species by the International Union for Conservation of Nature (IUCN), was found for the first time in the Port of Busan in 2017 and have been found a total of 11 times until 2020 [13]. *Anoplolepis gracilipes* was first found in the wooden packing material of freight entering the Port of Incheon from Vietnam in 2019 [14]. Second, alien species also imported deliberately for various purposes, such as pets and food with the development of transportation, which is considered as the major pathway of alien species in South Korea. Invasive alien species *Rana catesbeiana*, *Micropterus salmoides*, and *Lepomis macrochirus* were typically introduced for food, but now they are major invasive alien species that causes biodiversity decline. *Mauremys sinensis* and *Macrochelys temminckii* were widely raised as pets, but now they were banned from breeding, transfer, transportation, and importation because they cause great damage in the ecosystem, even though they are an endangered species as CITES [15].

In this study, we focused on evaluating the possibility of introduction of AAS, which is about the 1st step of biological invasion, through trade data among countries. To calculate the possibility of introduction of AAS, their origin and distribution of AAS were analyzed along with import data by country provided by the Korea Customs Service. Through this, implications for the current designation of AAS were proposed.

## 2. Materials and Methods

### 2.1. Species Distribution Database

As of April 2021, a total of 300 species have been designated as AAS in 10 taxonomic groups (Notification of the Ministry of Environment, No. 2020-79): mammals, birds, fish, mollusks, amphibians, reptiles, insects, spiders, other arthropods and plants. The data sources for natural and distribution range of each species are as follows: 1. Centre of Agriculture and Biosciences International Invasive Species Compendium (CABI ISC, https://www.cabi.org/isc, accessed on 1 July 2020); 2. Global Invasive Species Database (GISD, http://www.iucngisd.org/gisd/index.php, accessed on 1 August 2020); 3. IUCN Red List (https:// www.iucnredlist.org, accessed on 1 August 2020); 4. Catalogue of Life (http://www.catalogueoflife.org, accessed on 1 August 2020); 5. World Spider Catalog (https://wsc.nmbe.ch, accessed on 1 September 2020); and 6. Other data sources of reported regions invaded by alien species identified by an Internet search.

Based on import and export trade statistics from the Korea Customs Service (https://unipass.customs.go.kr/ets/index.do, accessed on 1 April 2021), trading countries

were classified into eight regions: Asia, Middle East, Europe, North America, Latin America, Africa, Oceania, and Islands of Oceania. The total number of trading countries included was 255, including 32 countries in Asia, 21 countries in Middle East, 58 countries in Europe, 2 countries in North America, 54 countries in Latin America, 56 countries in Africa, 18 countries in Oceania, and 14 countries in Islands of Oceania. When a species is distributed across a continent, it indicates that the species is distributed in a certain country belonging to the continent, not the entire continent.

### 2.2. Import Data

The Korea Customs Service provides data on the number of imports and weight imported from a port or airport. The import data required for analysis were collected through the following three ways for each trading country:

In order to compare the possibility of introduction of alien species including AAS through import itself, data on the number of imports for all import items for the 5 years from 2016 to 2020 were obtained. In addition, the number of imports for the 20 years from November 2001 to November 2020 was used to compare long-term changes in the number of imports by continent.

Data on the import weight related to living organisms were organized to evaluate the possibility of alien species directly imported or unintentionally mixed with other living organisms. The number of imports by item type was not recorded in the statistics; thus, the overall weight of import was used. First, import items related to living organisms were selected from "property classification" of the Korean Customs Service (Table 1). Then, import weight not related to living organisms in each item such as byproducts of processed items. However, the weight of Shellfish and Squids was recorded by a combination of live and refrigerated. The weight of imported items from trading countries from 2016 to 2020 was aggregated by continent. Because no items related to living organisms were imported from Island of Oceania, this region was excluded from the statistical evaluation by continent.

**Table 1.** Items used for the possibility of introduction by import of living organisms among "property classification" of the Korean Customs Service.

| Main Category | Subcategory | | Item | Classification |
|---|---|---|---|---|
| Consumer goods | Direct consumption goods | Agricultural products | All items except for products, by-products, and processed goods | Plants |
| | | Animal products | Live animals | Animals |
| | | Marine products | Fish (live fish) | |
| | | | Crustaceans (not smoked) | |
| | | | Shellfish (live, fresh, refrigerated) | |
| | | | Squids (live, fresh, refrigerated) | |
| | | | Seaweeds | |
| | | | Other marine products | |
| Raw materials | Fuel/raw materials of animals and plants | Agricultural products | Live part of trees | Plants |
| | | | Tree seeds | |
| | | | Flowering plants | |
| | | | Vegetables and their seeds | |
| | | | Feedstuff | |

The weight of imported AAS was organized based on the 23 Harmonized System (HS) codes (Table 2). The Ministry of Environment designated 23 HS codes which are highly likely to be related to the import of AAS. Though the weight was not suitable for comparison between animals and plants or between species, the number of import cases by HS codes were not recorded in the Korean Custom Service. The 23 HS codes were classified as mammals, reptiles, birds, insects, amphibians, fish, other arthropods, mollusks, other animals, and plants based on the properties of the items. Among the 23 HS codes,

"animal products (0410009000)" were excluded because they are not imported alive. "Other living organisms (0106909000)" were classified as "other animals" because there were several animal taxa in that code. Spiders of AAS were not included among the 23 HS codes. The weight of imported items by country from 2016 to 2020 was obtained for each item and aggregated by continent.

**Table 2.** Items subject to 23 HS Code related to AAS. No. 20, animal products were excluded from statistics because they were not imported alive.

| No. | HS CODE | Item | Classification |
|---|---|---|---|
| 1 | 0103920000 | Living organisms weighing 50 kg or more, such as pigs | Mammals |
| 2 | 0106149000 | Other species such as rabbits | Mammals |
| 3 | 0106193000 | Deer | Mammals |
| 4 | 0106196090 | Other species such as mink | Mammals |
| 5 | 0106199000 | Other mammals | Mammals |
| 6 | 0106201000 | Snakes | Reptiles |
| 7 | 0106203000 | Turtles | Reptiles |
| 8 | 0106209000 | Other reptiles | Reptiles |
| 9 | 0106390000 | Other birds | Birds |
| 10 | 0106490000 | Other insects | Insects |
| 11 | 0106901000 | Amphibians | Amphibians |
| 12 | 0106909000 | Other living organisms | Other animals |
| 13 | 0301119000 | Live ornamental fish | Fish |
| 14 | 0301911000 | *Salmo trutta*, etc. | Fish |
| 15 | 0301930000 | Carp | Fish |
| 16 | 0301999070 | Mudfish | Fish |
| 17 | 0301999080 | Catfish | Fish |
| 18 | 0306390000 | Crustaceans | Other arthropods |
| 19 | 0307310000 | Living organisms such as mussels, etc. | Mollusks |
| 20 | 0410009000 | Animal products | - |
| 21 | 0602909090 | Plants | Plants |
| 22 | 1209300000 | Flower seeds | Plants |
| 23 | 1209999000 | Sowing seeds | Plants |

*2.3. Calculation of the Possibility of Introduction*

To calculate the possibility of introduction ($P_i$), the number of AAS distributed in one country ($S_i$) was multiplied by the number of import or weight ($V_i$) from that country to South Korea. The number of AAS ($S_i$) and the number of import or weight ($V_i$) were divided by the largest value and ranged from 0 to 1, respectively.

$$P_i = S_i \times V_i$$

$$(S_1 = S_2 = S_5)$$

The possibility of introduction was calculated on the premise that the more AAS distributed in the country, or the more items and weight imported, the higher the possibility of introduction [9,12,16–18]. The possibilities of introduction from each country were aggregated for comparison by continent. A total of five possibilities of introduction ($P_i$) were calculated according to the taxanomic groups of AAS and types of import items (Table 3): $P_1$ simply represent the possibility of introduction from both the number of AAS in a country and the number of imports from the country based on the premise. $P_2$ indicates the possibility of AAS being deliberately introduced alive or introduced though a mixture with other living organisms. $P_3$ and $P_4$ indicate the possibilities of introduction of AAS

as items related to animal and marine species alive, respectively. Finally, $P_5$ indicates the possibility of AAS being introduced via items included in the 23 HS codes.

**Table 3.** Description of variables related to the possibility of introduction.

| $i$ | $P_i$ | $S_i$ | $V_i$ |
|---|---|---|---|
| 1 | Possibility of introduction by import | No. of AAS in one country | No. of import |
| 2 | Possibility of introduction by import of living organisms | No. of AAS in one country | Weight of imported living organisms (ton) |
| 3 | Possibility of introduction by import of animal products | No. of animals except fish among AAS in one country | Weight of imported animal species (ton) |
| 4 | Possibility of introduction by import of marine products | No. of fish among AAS in one country | Weight of imported marine species (ton) |
| 5 | Possibility of introduction by import of 23 HS codes | No. of AAS in one country | Weight of 23 HS codes-related imported AAS (kg) |

### 3. Results

*3.1. Distribution of AAS*

The results of investigating the natural range of AAS by continent showed that 213 of the 300 species had one continent as the natural range and the other 87 species had at least two continents. The number of species with a natural range in Asia was the highest with 106 species, followed by 85 species from Europe, 68 species from North America, 63 species from Latin America, 45 species from Africa, 29 species from Middle East, 23 species from Oceania, and no species from Islands of Oceania (Figure 1a). In terms of distribution, 69 species were distributed in a single continent, whereas the others were distributed throughout various continents. The majority of AAS were distributed in Asia with 199 species, followed by North America with 181 species, Europe with 172 species, Latin America with 150 species, Oceania with 117 species, Middle East with 113 species, Africa with 102 species, and Islands of Oceania with 41 species (Figure 1b).

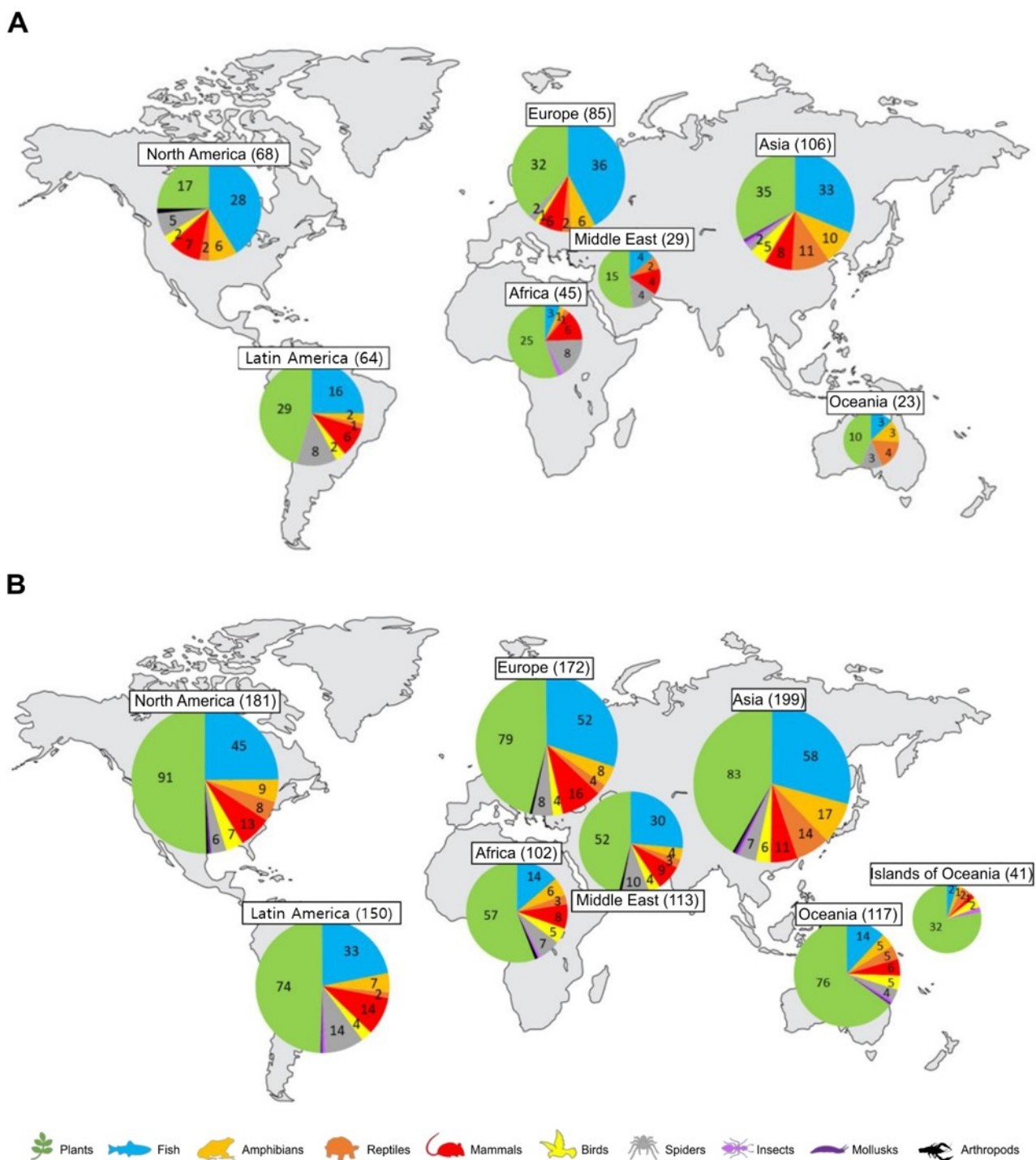

**Figure 1.** Number of alert alien species according to the natural range and distribution range (both natural range and alien range) by continent. (**A**) Natural range. (**B**) Distribution range.

When compared by taxa, the amphibians and reptiles designated AAS were mainly distributed in Asia. Similarly, fish mainly distributed in Asia and Europe were designated as AAS (Figure 1b). In the case of plants designated as AAS, when the natural range and distribution range were compared, the number was very large, so it was clearly seen that they were distributed as alien species in most continents. In particular, in the case of Islands of Oceania, all taxa designated as AAS were found to be invasive species. Of the 300 species of AAS, 17 species (1 species of birds, 2 species of fish, and 14 species of plants) were distributed in all eight continents.

Regarding the number of AAS distributed by country, the United States predominated with 179 out of 300 species (Table 4, Table S1), followed by China with 110 species, Australia with 101 species, France with 96 species, and Spain with 94 species. Fish were distributed mostly in the United States (43 species) and Russia (40 species) in comparison with other countries (<30 species). In the case of plants, 90 out of 99 species were distributed in the United States, followed by Australia with 71 species.

**Table 4.** The top 10 countries with large numbers of AAS distributed by taxonomic groups. Taxonomic groups with small numbers of species were excluded (birds, mollusks, insects and other arthropods). See Table S1 for the entire country rankings and the numbers of AAS distributed by countries in all taxonomic groups.

|  | Mammals (25) | Reptiles (22) | Amphibians (28) | Fish (84) | Spiders (32) | Plants (99) |
|---|---|---|---|---|---|---|
| 1 | USA (13) | China (9) | USA (9) | USA (43) | USA (7) | USA (90) |
| 2 | Mexico (10) | USA (8) | Japan(9) | Russia (40) | Argentina (6) | Australia (71) |
| 3 | Belgium (8) | Vietnam (8) | China (7) | Rumania (28) | South Africa (4) | Spain (57) |
| 4 | Italia (8) | Thailand (8) | Spain (6) | Canada (28) | Madagascar (4) | China (54) |
| 5 | The Czech Republic (8) | Myanmar (7) | UK (6) | Germany (27) | Israel (4) | India (50) |
| 6 | Croatia (8) | Bangladeshi (7) | France (6) | Bulgaria (27) | India (4) | France (50) |
| 7 | France (8) | India (7) | Taiwan (5) | France (26) | Canada (4) | Mexico (47) |
| 8 | Belarus (7) | Indonesia (7) | Denmark (5) | Ukraine (25) | Australia (4) | Argentina (43) |
| 9 | Slovakia (7) | Laos (6) | Germany (5) | China (25) | Russia (3) | Italia (42) |
| 10 | Austria (7) | Malaysia (6) | Mexico (5) | Kazakhstan (25) | United Arab Emirates (3) | New Zealand (41) |

### 3.2. Import Data

Asia recorded the largest number of imports, with 1,830,779 cases in 2001 increasing to 4,311,247 cases in 2010 and to 7,593,158 cases in 2020, indicating a 2.35-fold and 4-fold increase, respectively (Figure 2). North America recorded 1,000,000 cases in 2008 and 16,153,841 in 2020, showing the fastest growth rate. As of 2020, North America, Asia, Europe, and Oceania exceeded 1,000,000 cases of import. Among them, the United States had the most cases with 15,901,241 cases in a single year (2020), followed by China with 4,565,009 cases, Germany with 2,669,119 cases, and Japan with 1,864,174 cases (Table S1).

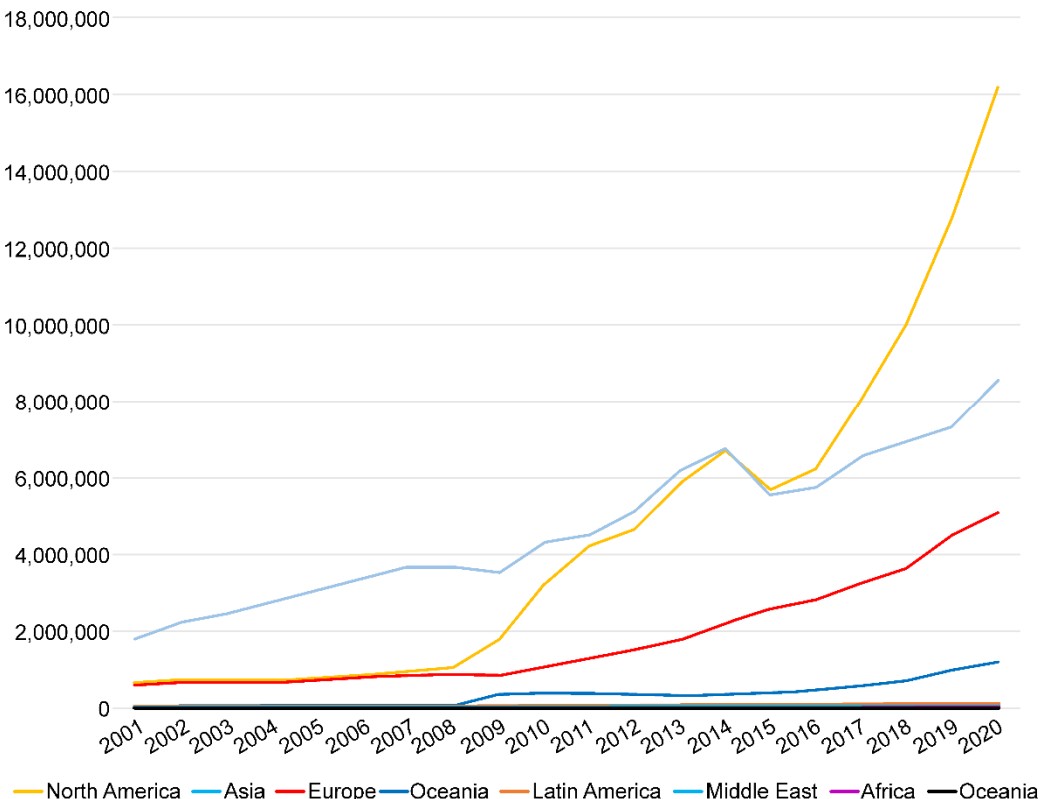

**Figure 2.** The total number of imports by continent.

Among the 255 trading countries, 157 countries were engaged in trade related to living organisms (Table 5). The weight of imported living organisms over the last 5 years totaled 154,782,190 tons from seven continents (excluding Islands of Oceania), of which 99.8% accounted for agricultural products. Approximately 41% of the total weight of imported agricultural products was from North America, followed by Latin America with approximately 20%. The largest import volume by country was, in order, the US, Brazil, Australia, China, and Argentina (Table 6). The amount of agricultural products imported from Asia was the largest at 43.5%, followed by North America at 28%. Marine products were also imported mostly from Asia at 57.5%, followed by Europe at 25.2%.

**Table 5.** The weight (in tons) of imported living organisms by continent.

|  | The Total Number of Trading Countries | The Number of Relevant Countries | Total Weight | Agricultural Products | Animal Products | Marine Products |
|---|---|---|---|---|---|---|
| Asia | 32 | 27 | 22,132,038 (14.30%) | 21,932,304 (14.20%) | 3265 (43.50%) | 196,469 (57.52%) |
| North America | 2 | 2 | 63,442,312 (40.99%) | 63,427,137 (41.07%) | 2106 (28.06%) | 13,069 (3.83%) |
| Latin America | 54 | 29 | 30,823,371 (19.91%) | 30,804,156 (19.95%) | 189 (2.52%) | 19,027 (5.57%) |
| Europe | 58 | 40 | 21,664,536 (14.00%) | 21,576,766 (13.97%) | 1798 (23.96%) | 86,001 (25.18%) |
| Oceania | 14 | 7 | 15,077,894 (9.74%) | 15,077,620 (9.76%) | 136 (1.81%) | 139 (0.04%) |
| Africa | 56 | 36 | 1,572,800 (1.02%) | 1,569,256 (1.02%) | 10 (0.13%) | 3,543 (1.04%) |
| Middle East | 21 | 16 | 69,239 | 45,921 | 2 | 23,316 |

| | | | | |
|---|---|---|---|---|
| | | (0.04%) | (0.03%) | (0.03%) | (6.83%) |
| Islands of Oceania | 18 | - | - | - | - | - |
| Total | 255 | 157 | 154,782,190 (100%) | 154,433,158 (100%) | 7506 (100%) | 341,564 (100%) |

**Table 6.** The order of continents with the large weight of imported living organisms.

| | Total Sum | | Agricultural Products | | | Animal Products | | | Marine Products | | |
|---|---|---|---|---|---|---|---|---|---|---|---|
| Order | Country | Weight (ton) | Order | Country | Weight (ton) | Order | Country | Weight (ton) | Order | Country | Weight (ton) |
| 1 | USA | 60,331,124 | 1 | USA | 60,316,747 | 1 | China | 2896 | 1 | Vietnam | 171,554 |
| 2 | Brazil | 16,629,299 | 2 | Brazil | 16,629,275 | 2 | USA | 1308 | 2 | Russian Federation | 80,531 |
| 3 | Australia | 14,522,032 | 3 | Australia | 14,521,950 | 3 | Netherlands | 1036 | 3 | Malaysia | 19,745 |
| 4 | China | 11,804,020 | 4 | China | 11,801,124 | 4 | Canada | 798 | 4 | Bahrain | 13,168 |
| 5 | Argentina | 10,263,207 | 5 | Argentina | 10,253,066 | 5 | Denmark | 191 | 5 | USA | 13,069 |
| 6 | Ukraine | 7,574,825 | 6 | Ukraine | 7,574,824 | 6 | France | 188 | 6 | Argentina | 10,132 |
| 7 | Russian Federation | 6,873,080 | 7 | Russian Federation | 6,792,538 | 7 | Bahamas | 178 | 7 | Mexico | 7826 |
| 8 | Philippines | 3,780,228 | 8 | Philippines | 3,780,225 | 8 | Germany | 169 | 8 | Saudi Arabia | 6166 |
| 9 | Vietnam | 3,352,726 | 9 | Serbia | 3,225,854 | 9 | Japan | 168 | 9 | Norway | 4373 |
| 10 | Serbia | 3,225,855 | 10 | Vietnam | 3,181,165 | 10 | Indonesia | 130 | 10 | Morocco | 3970 |
| 11 | Canada | 3,111,188 | 11 | Canada | 3,110,391 | 11 | Australia | 82 | 11 | Myanmar | 3629 |
| 12 | Paraguay | 1,883,921 | 12 | Paraguay | 1,883,921 | 12 | Spain | 55 | 12 | Namibia | 2709 |
| 13 | Thailand | 1,875,191 | 13 | Thailand | 1,875,142 | 13 | New Zealand | 53 | 13 | Taiwan | 821 |

According to the import weight data arranged by the 23 HS codes, a total of 70,412,084 kg was imported over the last 5 years from seven continents, and the weight of imported living organisms (154,782,227 tons) was extremely small at 0.05% (Table 7). Asia accounted for the largest weight at 94.32%. According to the taxonomic groups, fish comprised 76.66% of the entire weight of import, and plants comprised 22.11%, indicating that both groups accounted for 98.77% of all import weight.

**Table 7.** The order of continents with the large import weight (kg) of 23 HS codes by taxonomic groups of AAS.

| | Continent | Total Weight | Mammals | Reptiles | Birds | Insects | Amphibians | Fish | Other Arthropods | Mollusks | Plants | Other Animals | |
|---|---|---|---|---|---|---|---|---|---|---|---|---|---|
| 1 | Asia | 66,415,365 (94.324%) | 226,281 (0.34%) | 39,823 (0.06%) | 110 (0.00%) | 41 (0.00%) | 99,430 (0.15%) | 53,957,360 (81.24%) | 3345 (0.01%) | | 12,088,971 (18.20%) | 4 (0.00%) | (100%) |
| 2 | North America | 1,624,946 (2.308%) | 41,564 (2.56%) | 6256 (0.38%) | 98 (0.01%) | 593 (0.04%) | 1108 (0.07%) | 215 (0.01%) | | 31 (0.00%) | 1,574,990 (96.93%) | 91 (0.00%) | (100%) |
| 3 | Europe | 952,508 (1.353%) | 38,966 (4.09%) | 4257 (0.45%) | 1633 (0.17%) | 144,931 (15.22%) | 297 (0.03%) | 3973 (0.42%) | 1 (0.00%) | 174,118 (18.28%) | 519,691 (54.56%) | 64,641 (6.79%) | (100%) |
| 4 | Africa | 9890 (0.014%) | 715 (7.23%) | 5164 (52.21%) | 22 (0.22%) | | 278 (2.81%) | 941 (9.51%) | | | 2756 (27.87%) | 14 (0.14%) | (100%) |
| 5 | Oceania | 9908 (0.014%) | 1755 (17.71%) | 14 (0.14%) | | 13 (0.13%) | | 137 (1.38%) | 171 (1.73%) | 7445 (75.14%) | 368 (3.71%) | 5 (0.05%) | (100%) |
| 6 | Lain America | 1,397,362 (1.985%) | 576 (0.04%) | 581 (0.04%) | 20 (0.00%) | | 28 (0.00%) | 15,900 (1.14%) | | | 1,380,257 (98.78%) | | (100%) |
| 7 | Middle East | 2105 (0.003%) | 504 (23.94%) | 1416 (67.27%) | | 3 (0.14%) | 19 (0.90%) | 120 (5.70%) | | | 36 (1.71%) | 7 (0.33%) | (100%) |
| | Total | 70,412,084 (100%) | 310,361 (0.44%) | 57,511 (0.08%) | 1883 (0.00%) | 145,581 (0.21%) | 101,160 (0.14%) | 53,978,646 (79.66%) | 3517 (0.00%) | 181,594 (0.26%) | 15,567,069 (22.11%) | 64,762 (0.09%) | (100%) |

In terms of the imported AAS by taxonomic group, China had the largest import weight of mammals, reptiles, amphibians, fish, and plants, accounting for 90.86% overall

(Table 8). Mammals were mostly imported from Japan, China, and Denmark. Reptiles were mostly imported from the US, China, and Nicaragua, whereas birds and insects were mostly imported from the Netherlands. For amphibians, China predominated at 97%, followed by the US and Indonesia. China also predominated for fish at 63.7%, followed by the US, Indonesia, and Costa Rica. For other animals, the Netherlands was the largest importing country accounting for 74.2% of weight.

**Table 8.** The order of countries with the large weight (Kg) of 23 HS codes by taxonomic groups of AAS.

| | Mammals | Reptiles | Birds | Insects | Amphibians | Fish | Other Arthropods | Mollusks | Plants | Other Animals |
|---|---|---|---|---|---|---|---|---|---|---|
| 1 | Japan (149,950) | USA (32,343) | Netherlands (946) | Netherlands (134,029) | China (98,138) | China (53,838,992) | Indonesia (1990) | Russia (174,118) | China (9,930,594) | Netherlands (48,062) |
| 2 | China (73,420) | China (5352) | Spain (320) | Belgium (6700) | USA (947) | Indonesia (52,792) | Thailand (469) | New Zealand (7445) | USA (1,558,018) | Belgium (8619) |
| 3 | Denmark (23,780) | Nicaragua (3702) | Germany (210) | Spain (3826) | Indonesia (691) | Sri Lanka (30,069) | China (351) | Canada (31) | Indonesia (1,553,410) | Spain (7701) |
| 4 | USA (21,717) | Peru (2856) | Czech (100) | USA (560) | Hong Kong (208) | Singapore (25,050) | Singapore (282) | | Costa Rica (1,329,013) | Germany (250) |
| 5 | Canada (19,847) | Venezuela (1977) | USA (88) | Austria (171) | Togo (206) | Columbia (10,149) | Taiwan (252) | | Netherlands (415,894) | USA (90) |
| 6 | Netherlands (5118) | Syria (1912) | Philippines (58) | Turkey (130) | Netherlands (161) | Taiwan (6633) | Australia (171) | | Thailand (288,593) | UK (9) |
| 7 | Germany (2343) | Ghana (1909) | Japan (52) | Switzerland (39) | Canada (161) | Peru (5656) | Germany (1) | | Philippines (109,801) | Togo (8) |
| 8 | France (2103) | Mauritius (904) | Portugal (40) | Canada (33) | Taiwan (120) | Japan (2662) | Sri Lanka (1) | | Taiwan (73,855) | Egypt (7) |
| 9 | Russia (1860) | Jordan (634) | Tanzania (22) | Germany (25) | Japan (103) | Germany (2428) | | | Guatemala (43,811) | Australia (5) |
| 10 | Australia (1662) | Belize (557) | Argentina (20) | Japan (22) | Singapore (100) | Czech (1393) | | | Denmark (39,513) | Ghana (2) |
| | Other countries (8561) | Other countries (5365) | Other countries (27) | Other countries (46) | Other countries (325) | Other Countries (2822) | | | Other countries (244,567) | Other countries (9) |
| Total | 310,361 | 57,511 | 1883 | 145,581 | 101,160 | 53,978,646 | 3517 | 181,594 | 15,567,069 | 64,762 |

*3.3. The Possibility of Introduction*

The results of comparing the possibility of introduction by import ($P_1$) by country showed that the US (1.000) had the highest value compared to other countries, followed by China (0.212), Germany (0.075), Japan (0.070), and Italy (0.025) (Table 9). Comparing $P_1$ by continent, North America (1.0060) had the highest value, followed by Asia (0.3269) and Europe (0.1435) (Table 10). The results of comparing the possibility of introduction by import of living organisms ($P_2$), instead of total number of imports, showed the US (1.000) to have the highest value, followed by Australia (0.136), China (0.120), Brazil (0.086), and Argentina (0.060). Comparing $P_2$ by continent, North America (1.0213) had the highest value like that in $P_1$, followed by Asia (0.1689) and Latin America (0.1593). Excluding agricultural products, which accounts for 99.8% of the import of living organisms, the results of the possibility of introduction by import of animal species ($P_3$) showed that China (0.705) had the highest value, followed by the US (0.452), Canada (0.106), Netherlands (0.090), and Japan (0.037). Regarding marine species ($P_4$), Russia (0.417) had the highest value, followed by Vietnam (0.200), the US (0.076), Malaysia (0.026), and Mexico (0.024). Lastly, calculating the possibility of introduction by import of 23 HS codes ($P_5$) revealed the highest value for China (0.615), followed by the US (0.025), Indonesia (0.007), Costa

Rica (0.005), and Netherlands (0.003); the results by continent were Asia (0.6256), North America (0.0251), and Europe (0.0052).

**Table 9.** The order of countries and continents with the high possibility of introduction of AAS.

| Country Order | $P_1$ | $P_2$ | $P_3$ | $P_4$ | $P_5$ |
|---|---|---|---|---|---|
| 1 | USA (1.000) | USA (1.000) | China (0.704) | Russia (0.417) | China (0.615) |
| 2 | China (0.212) | Australia (0.136) | USA (0.452) | Vietnam (0.200) | USA (0.025) |
| 3 | Germany (0.075) | China (0.120) | Canada (0.106) | USA (0.076) | Indonesia (0.007) |
| 4 | Japan (0.070) | Brazil (0.086) | Netherlands (0.090) | Malaysia (0.026) | Costa Rika (0.005) |
| 5 | Italy (0.025) | Argentina (0.090) | Japan (0.037) | Mexico (0.024) | Netherlands (0.003) |
| 6 | Australia (0.023) | Russia (0.052) | France (0.028) | Norway (0.010) | Thailand (0.001) |
| 7 | UK (0.013) | Ukraine (0.043) | Indonesia (0.021) | Argentina (0.008) | Russia (0.001) |
| 8 | New Zealand (0.012) | Canada (0.021) | Germany (0.019) | Morocco (0.006) | Japan (0.001) |
| 9 | France (0.011) | Vietnam (0.016) | Denmark (0.018) | Myanmar (0.003) | Philippines (0.0004) |
| 10 | Vietnam (0.011) | Philippines (0.016) | Australia (0.014) | Saudi Arabia (0.002) | Taiwan (0.0003) |

**Table 10.** The order of continents with the high possibility of introduction of AAS.

| Continent Order | $P_1$ | $P_2$ | $P_3$ | $P_4$ | $P_5$ |
|---|---|---|---|---|---|
| 1 | North America (1.0060) | North America (1.0213) | Asia (0.7736) | Europe (0.4298) | Asia (0.6256) |
| 2 | Asia (0.3269) | Asia (0.1689) | North America (0.5582) | Asia (0.2301) | North America (0.0251) |
| 3 | Europe (0.1435) | Latin America (0.1593) | Europe (0.1784) | North America (0.0762) | Europe (0.0052) |
| 4 | Oceania (0.0354) | Oceania (0.1388) | Oceania (0.0196) | Latin America (0.0327) | Latin America (0.0051) |
| 5 | Latin America (0.0059) | Europe (0.1318) | Latin America (0.0081) | Middle East (0.0081) | Oceania (0.0001) |
| 6 | Middle East (0.0009) | Africa (0.0072) | Africa (0.0003) | Africa (0.0006) | Africa (0.00003) |
| 7 | Africa (0.0005) | Middle East (0.0002) | Middle East (0.0001) | Oceania (0.0001) | Middle East (0.000005) |

## 4. Discussion

According to our analysis of the natural range and the distribution of AAS, the largest number of species was native to Asia. The results indicated that Asia was the natural range of 106 species and the distribution of 199 out of the 300 species. In contrast, in the comparison by country, the AAS were distributed in the US, China, Australia, France, and Spain in order. Notably, 90 out of 99 plant species designated as AAS were distributed in the US. Plants had the widest geographical distribution because they are more convenient to store and transport compared with animals, which have many constraints for live transport, and have a longer duration of survival.

However, the possibility of introduction considered together with distribution and import data showed different results from the distribution alone. First, the possibility of introduction by import ($P_1$), calculated together with the number of import and the distribution data, was equivalent to the countries with the largest number of imports in the order of the US, China, Germany, Japan, and Italy. As in the case of Spain and France, if there were a low number of imports, despite the AAS being distributed more than in other countries, the possibility of introduction was much lower. Several studies have investigated the relationship between the introduction of alien species and trade. Liebhold et al. [19] used invasion history records to show that the freight of passengers entering the US was an important introduction channel for alien insects. Moreover, their results showed that the number of alien insects whose introduction blocked was relative to the volume of air transportation entering the country. Tatem et al. [20] found that the volume of marine

transportation and climate played major roles in the dispersal of the Asian tiger mosquito (*Aedes albopictus*), which spreads dengue. Westphal et al. [12] conducted regression tree analyses of the distribution of alien species belonging to the GISD using a total of 26 variables and found that the degree of international trade was the most accurate variable predicting the number of alien species in one country.

Even though $P_1$ corresponds to the entire possibility of introduction through various routes, including unintentional introduction by trade itself, biological invasion is closely related to directly importing live alien species [17,21]. In South Korea, there have been many past cases of unintentional results of such imports, including the spread of infectious diseases, introduction of plant diseases and insect pests, and threatened ecosystems [22]. In this study, the possibility of introduction by import of living organisms ($P_2$) was different from $P_1$ calculated using the total number of import cases. Although Germany, Japan, and Italy had a large number of imports in $P_1$, they showed a low possibility of introduction related to living organisms. $P_2$ was high for the US, Australia, China, Brazil, and Argentina in that order. Furthermore, the low value in Brazil is due to the small number of distributed AAS despite the high weight of imports. When animal species were calculated separately from living organisms, the possibility of introduction of animal species ($P_3$) was high for the US, Canada, Netherlands, and Japan. When marine species were calculated separately from living organisms, Russia, Vietnam, the US, Malaysia, and Mexico had a high possibility of introduction ($P_4$) value. The results of calculating countries with a high possibility of introduction according to certain imported items showed different results. Thus, several studies have specifically compared the import volume of related items to investigate the possibility of introduction of particular taxonomic groups. Chapman et al. [17] found that when countries belonging to the Europe and Mediterranean Plant Protection Organization increased the volume of imports of agricultural products, invasive plant pests (invertebrates, pathogens, and plants) also increased. Similarly, Bradie et al. [23] predicted invasive alien fish with high possibility of settlement using import records of live fish and data on ornamental fish.

More specifically, the import weight of items that are highly likely to be related to import of AAS was inferred from the weight of imported items corresponding to the 23 HS codes. In the comparison by continent, Asia accounted for 94.32% of overall import weight. However, in the comparison by country, Indonesia, Costa Rica, Netherlands, and Thailand, where few AAS are distributed, had a large import weight of AAS, thus indicating a high possibility of introduction ($P_5$). For the taxonomic groups, there were large differences between the number of AAS distributed in a country and the import weight of items related to the import AAS from the country.

Since the possibility of introduction of alien species is higher in countries with a large number of import cases or import weight, it is appropriate to preferentially manage invasive alien species distributed in that countries as AAS. However, when looking at the relationship between the distribution of AAS and the trade, it seems that these points were not taken into account when designating AAS. In the case of Fish, which account for 76% of the import weight of AAS, 43 and 40 species of the 84 species of AAS were distributed in US and Russia, respectively. However, the import weight of items related to import of fish designated in AAS from these countries were extremely low. Although 99.7% of these items were imported from China, only 25 species were distributed in the United States, but the import weight related to the import of plants designated as AAS accounts for only 10% of the total. Although Australia and Spain showed extremely low import weight of plants, 71 and 57 of the alert alien plant species, respectively, were distributed in these counties, which is higher than other countries.

These predictive results may be influenced by the diversity and accuracy of variables included in the calculation. In order for alien species to successfully invade a new habitat, the propagule pressure, understood as a composite measure of the number of individuals released into a region in which they are not native, is important, which is estimated using the import weight and the number of imports in this study [10,23,24]. The increased cases

and import weight can overwhelm the impact of species characteristics on biological invasion because they reduce the effect of genetic bottlenecks and increase the chances of genetic variation that are favorable in the invaded places [25,26]. In this study, which deals with several taxa, the possibility of introduction of AAS was calculated using only the import weight and distribution of AAS because it is difficult to reflect the characteristics of each taxonomic group with respect to biological invasion. However, according to the "tens rule," which states that only 10% of introduced alien species become invasive through establishment and dispersal, habitat suitability cannot be disregarded [27]. In order to effectively manage alien species with limited cost, it is very important to prioritize among the introduced alien species. As a result of analyzing the two main stages of biological invasion, introduction and establishment, across the world, habitat suitability must be reflected in information on introduction pathways in order to identify the area or species that should be managed first [28,29]. In a study by Chapman et al. [17] of various models predicting the possibility of biological invasion, the most accurate model considered both climate similarity and trade volume. Thus, future research assessing the possibility of introduction of AAS should consider the species-specific life cycle of each AAS and properties of habitats, including climate along with the possibility of introduction by import.

## 5. Conclusions

As globalization progresses, human activities, such as travel and trade, are rapidly increasing beyond national boundaries. It is increasingly recognized that places, such as ports and airports, where trade occurs play a major role as introduction pathways for alien species. Owing to such human activities, the number of alien species are increasing worldwide [30,31]. Accordingly, South Korea has started to manage alien species through the designation of AAS which may harm ecosystems if introduced into South Korea. As of April 2021, 300 species have been designated as AAS from 10 taxonomic groups, including mammals, birds, fish, mollusks, amphibians, reptiles, insects, spiders, other arthropods and plants [7].

In this study, the natural range and distribution of AAS were analyzed along with import data by countries to calculate the possibility of introduction of AAS. In the process of designation of AAS to prevent introduction of invasive alien species, it seems that the close relationship between trade and introduction of alien species was not fully considered. For future management plans for non-introduced alien species, species with high possibility of introduction into South Korea through trade should be prioritized using import data. It is also necessary to study hitchhikers, which were known to move through trade and were considered a major introduction route for alien species, but were not included in the trade data. Moreover, in order to prevent the introduction of AAS and calculate more accurate possibility, a more efficient quarantine system than the current 23 HS codes is needed. To that end, it is essential that related authorities work collaboratively, including the Ministry of Environment, Korea Customs Service, and Animal and Plant Quarantine Agency.

There are many papers dealing with introduction risks of alien species. However, most studied focused on the current status, spread and establishment of alien species already-established rather than how alien species will be introduced [16,32,33]. There is few research comparing the possibility of introduction of AAS that should be prevented with top priority among non-introduced alien species in South Korea. Unlike other studies comparing overall trade volume and the number of trade cases, the import items related to living organisms was selected and the possibility of introduction was compared in detail in this study. This approach is meaningful in determining the priority for the management and prevention of invasive alien species by obtaining predictive values of the introduction, which is the first step of biological invasion. We hope that this study can serve as a stepping stone for more accurately predicting the possibility of introduction of AAS in the future.

**Supplementary Materials:** The following supporting information can be downloaded at: https://www.mdpi.com/article/10.3390/d14110910/s1, Table S1: Total data of number of AAS and number of import /weight of each country.

**Author Contributions:** Conceptualization, A.J. and D.K.; data curation, A.J. and S.S.; formal analysis, A.J. and S.S.; funding acquisition, D.K.; investigation, A.J. and S.S.; methodology, A.J.; project administration, D.K.; supervision, D.K.; visualization, A.J.; writing—original draft, A.J.; writing—review and editing, A.J. and D.K. All authors have read and agreed to the published version of the manuscript.

**Funding:** This research was funded by the National Institute of Ecology (NIE), funded by the Ministry of Environment (MOE) of the Republic of Korea (NIE-A-2022-08 and NIE-D-2022-09).

**Institutional Review Board Statement:** Not applicable.

**Data Availability Statement:** Not applicable.

**Conflicts of Interest:** The authors declare no conflicts of interest. The funders had no role in the design of the study; in the collection, analyses, or interpretation of data; in the writing of the manuscript; or in the decision to publish the results.

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
