# Peer review of "Using Import Data to Predict the Potential of Introduction of Alert Alien Species to South Korea"

_diversity, doi:10.3390/d14110910_

Round 1

Reviewer 1 Report

This manuscript evaluated the introduction risks of a total of 300 alien species through trade from other countries across the globe. The authors have conducted intensive data collection on the alien species potentially introduced into South Korea. The topic also contributes to the development of early prevention strategies of alien species introduced into the country. However, I have three major concerns on the manuscript.

First, the Introduction section is needed to be focus more on the introduction risks and related quantification methods such as network analyses (e.g., Frost et al. 2019). Unfortunately, the authors provided too much invasion science background that is not closely related with introduction risks.

Second, regarding the method quantifying the introduction risks, it has been suggested that the probability of introduction risks in a certain country is highly correlated with the spatial distributions of human population density. Therefore, several previous studies have applied an introduction epicenter framework by accounting for the potential effect of human movement on alien species introduction risks (e.g., Early et al. 2016, Liu et al. 2019).

Third, the authors need clarify the current status of these introduced alien species in South Korea. There are differences in the probability of these alien species entering into the field and thus it needs to be incorporated this information to your predictive framework.

References used in the review and should be addressed in a revised version:

Early R, Bradley BA, Dukes JS, et al. 2016. Global threats from invasive alien species in the twenty-first century and national response capacities. Nat Commun 7: 12485.

Frost CM, Allen WJ, Courchamp F, et al. 2019. Using network theory to understand and predict biological invasions. Trends Ecol Evol 34: 831-843.

Liu X, Blackburn TM, Song T, et al. 2019. Risks of biological invasion on the belt and road. Curr Biol 29: 499-505.e4.

Reviewer 2 Report

The article is very interesting from a theoretical and practical point of view and provides original insights. 

Some editorial and discussion comments are given below:

1. It is inappropriate to write that after 1500 Europeans moved to the United States (line 25) and should be changed to North America. At that time, the United States of America did not exist as a country. 

2. Latin names of taxa should be italicized throughout the text.

3. In my opinion, the introduction should at least briefly discuss not only the legislation, but also existing knowledge on the introduction of alien species with various goods, with some specific examples.

4. The codes used in the text (lines 158-172) do not correspond to the codes in Table 3 (compare P_i and Pi). 

5. I cannot agree with the authors' use of the term 'place of origin'. The places of origin of many species of organisms differ significantly from their natural range. Place of origin and natural range are not identical concepts. I therefore suggest that the term 'native range' be used in the text instead of place of origin.  When dealing with secondary ranges, it is preferable to use the term 'alien range' or its equivalent.

6. The sentence (lines 192-194) uses a jargon-like expression, which may lead to a very different understanding of the statement. The sentence should be edited. 

7. I would suggest that the discussion should look at and discuss cases where endangered alien species have been introduced into the country with certain goods and have been recorded. This aspect is not touched upon in the discussion, although it would greatly contribute to highlighting the urgency and scale of the problem.

8. The authors state that "Reports on the initial introduction pathways and steps of introduction are scarce [15,29,30,31]". To what extent are such data scarce? Is it at country level or in general? There are a number of published studies on the introduction of alien species into new areas with imported commodities, especially cereals and other agricultural products. 

Round 2

Reviewer 1 Report

The manuscript has much been improved. However, I suggest that the authors need add some more discussions about the previous framework quantifying alien species invasion risks along different stages (e.g., Early et al. 2016, Liu et al. 2019). These important references and previous works have not been cited and discussed in the revised manuscript.

References in the review process and should be addressed in a revised version:

Early R, Bradley BA, Dukes JS, et al. 2016. Global threats from invasive alien species in the twenty-first century and national response capacities Nat Commun 7: 12485.

Liu X, Blackburn TM, Song T, et al. 2019. Risks of biological invasion on the belt and road Current Biology 29: 499-505.e4.

Reviewer 2 Report

The article has been significantly improved. I thank the authors for their detailed replies and explanations. A few minor comments on the text remain.

1. The group Arthropods is not properly defined (Fig. 1, Table 4, Table 7, etc.). Arthropods include insects, arachnids, crustaceans etc.  If other arthropods are to be defined, then they should be given a different name (Other arthropods). In addition, I would suggest changing the icon in Fig. 1 (Arthropods), which represents a scorpion. The scorpion is a member of the arthropods, but at the same time it belongs to the class of arachnids, which is listed separately.  

2. I suggest you read the text very carefully and pay attention to the use of terms. Very often, terms used in a particular context take on a different meaning. It is important that the text in scientific articles is clearly understandable and that there are no ambiguities that could lead to different interpretations. For example, the frequently used phrase "distributed range" should be replaced by "distribution range". Consideration should also be given to whether the phrase "imported cases" (Fig. 2) accurately reflects the message. Does it mean "cases of import" or "import volume" or "cases of alien species import"? 
